# Does Social Relation or Economic Interest Affect the Choice Behavior of Land Lease Agreement in China? Evidence from the Largest Wheat–Producing Henan Province

**Ruifeng Liu [1]**, **Zhifeng Gao [2]**, **Yefan Nian [2]** **and Hengyun Ma [1,\*]**

[1]   College of Economics and Management, Henan Agricultural University, Zhengzhou 450046, China; ruifeng076@163.com

[2]   Food and Resource Economics Department, University of Florida, P.O. Box 110240, Gainesville, FL 32611, USA; zfgao@ufl.edu (Z.G.); yfnian@ufl.edu (Y.N.)

\*   Correspondence: h.y.ma@163.com; Tel.: +86-371-5699-0018

**Abstract:** As China is transferring from a centrally planned economy to a market economy, the land market becomes more active. With the rapid transition from an agricultural society to an industrialized society, both economic and social factors could influence the choice of land lease agreement. This paper investigates the choice behavior of land lease agreement based on the household data collected from the largest wheat-producing province in China, Henan province. The results provide strong evidence that economic interest plays a key role in the choice of land lease agreement. Meanwhile, the paper has also found that social relations still affect the choice of land lease agreement. To speed up the transfer of land use rights and scale of operation, this study proposes: reducing the land dependence of rural households; improving agricultural machinery and achieving farming mechanization; creating more non-farm work opportunities for rural laborers; promoting the citizenization of rural households; respecting farmers' choice of contracts; and attaching importance to the role of social relations in farmland transfer.

**Keywords:** economic interests; social relations; land lease agreement; transfer of land use rights; China

## 1. Introduction

The Household Responsibility System (HRS) implemented by the Chinese government in the early 1980shas effectively improved Chinese farmers' enthusiasm for production and agricultural operation performance [1–4]. The 1978 HRS decollectivized China's agricultural production system by allocating farmers' 15-year land-use rights. The farmland was still collectively owned, but the households became residual claimants to their agricultural output [3]. Prior to the reforms of HRS, during 1952–1978, China's annual agricultural growth was sluggish at only 2.9%, where the annual growth rate of crops was 2.5% and grain was 2.4%. Under the HRS, however, China's agriculture sector grew at a respectable rate of 7.7% per year, where the annual growth rate of crops was 5.9% and grain was 4.8% during 1978–1984 [5]. In 1984, China's total grain output reached 407.31 million tons, an increase of 102.54 million tons over 1978. Therefore, the HRS was identified as the main source to improve total factor productivity and to account for about 46.89% of the output growth [3,6].

However, with the development of the commodity economy and large-scale commercialized production, the HRS has difficulty in meeting the needs of agricultural modernization and rural land scale operation due to the land fragmentation and small operation scale [7,8]. According to the National Rural Fixed Observation Point System [9], in 1986, on average each rural household owned

9.4 mu (1 mu ≈ 0.067 ha) of arable land, which were scattered across eight different plots. In 2009, each rural household had 7.1 mu arable land, scattered still across four different plots [10]. This land fragmentation not only deteriorates land use efficiency but also hinders the use of large agricultural mechanical application, increasing commuting and management costs [11]. Moreover, small-scale farming means that agricultural production surpluses are too small to meet basic household income needs [12].

Therefore, China has started to implement a series of policies and regulations to promote the transfer of rural land use rights [13,14]. These policies are aimed at reducing the efficiency loss caused by land fragmentation [15–17], scaling up land operations appropriately, and improving rural land use efficiency [13,18–20]. As a result, by the end of June 2016, the area of contracted rural land transfer nationwide reached 460 million mu, accounting for more than one third of the total area of cultivated land. Of China's 230 million rural households, more than 70 million have transferred their land, accounting for more than 30% of China's total rural households [21]. These statistics indicate that the scale of land transfer presents a good growth trend.

However, the current rural land rental market is still immature [22,23]. On the one hand, China's land market has a high non-participation rate of 70% [24], compared with 37% in Bangladesh [25] and 54% in India [26]. On the other hand, the nature of China's rural land transfer is highly informal. For example, most farmers rent/rent out the land from/to their relatives, friends, or fellow villagers for free [27,28]. There may be three reasons for this. First, the legal awareness of farmers is still weak, and they have insufficient understanding of formal and standardized land lease agreements [29]. Second, the geographical or genetic relationship between the two parties of the transfer land use rights is still very close [8]. Third, above all, it is hard to claim violation loss even if two parties sign a written land lease agreement. In other words, violation cost is very low, particularly in rural China. Therefore, it is not necessary to sign a written land lease agreement to bind both parties to a possible breach of contract.

Theoretically, informal land lease agreement arrangements (such as oral and the undefined term of tenancy) may have a negative impact on farmers' long-term land investment [18], while formal written land lease agreements can promote the farmer to enlarge the scale of production significantly [30]. Practically, most farmers who rent or rent out rural land rely only on verbal—rather than written—land lease agreements [1,6,18]. In fact, oral land lease agreements and "empty contracts" without any content clauses still accounted for 54.07% of land lease agreements by 2017 [8]. This raises the question of why farmers did not choose formal land lease agreements in their transfer of land use rights? Given the traditional rural relation society, does economic interest or social relation play a more important role in the choice of land use right transfer contracts?

Previous studies have found many factors influencing the choice of household land lease agreements. Those factors include incentives and risks [31], transaction attributes [32,33], the relationship between transaction parties [34,35], reputation [36,37], moral hazards [38], and property rights [39,40]. More recently, economists have started to explore the role of social relationship networks in farmers' land transfer. With the emergence, development, and evolution of new social economics, social networks are used to investigate economic activities. Economists increasingly appreciate the role social networks play in promoting employment, increasing wages, obtaining private loans, and adopting agricultural technologies [41–44]. They have found that, on the whole, the transfer of rural land occurs mainly among relatives, friends, neighbors, and acquaintances [1,45]. Lessors tend to prefer to rent out their land to lessees with a close relationship to them, and are willing to lower or even remove the rental price [18,35,46,47]. In addition, they investigated the efficiency of rural land transfers to relatives/non-relatives, but they found the results different. For example, Sadoulet et al. (1997) found that sharecroppers who had a kinship relationship with the landlord were more effective than non-kin sharecropping contracts in the optimal social investment and effort on their land [48]. Holden and Ghebru (2006) found that land rental between kin was just as effective as between non-kin [49], while Kassie and Holden (2007) argued that land rented by non-kin was more

productive than land rented by kin [50]. Furthermore, Kirwan (2009) found that a relationship between the landlord and tenant creates trust, which in turn affects rental rate [51]. Tang et al. (2019) also suggested that social ties based on kinship and geographical location increased rent deviations [28]. However, Bryan et al. (2015) found no strong evidence that landlord−tenant family relationships affected cash rents in southern Ontario, Canada [52].

The aim of this paper is to study choice behavior of rural land lease agreement and evaluate the impacts of both economic interests and social relations. The paper contributes to the nascent literature in threefold. First, we believe that major land lease agreement clauses (i.e., rental period, transfer scale and rental fee) may affect farmers' choice of land lease agreement type. This empirical assumption differs from that of previous studies that argue that in practice, decisions on major clauses of land lease agreement are often taken together with, or even after, the decision about using a written or oral land lease agreement [53–55]. Second, we use different contracting partners as key explanatory variables to examine the impact of rural social relations on the selection behavior of land lease agreement. Empirically, we employ the theory of differential mode of association (*Chaxugeju*) proposed by Fei in 1948 to fill the gap that the current studies are less concerned with the uniqueness and differences of the transfer of rural land use rights in the context of Chinese culture [56]. Third, we consider the possibility that the proportion and the scale of the transferred land may have different influences on the choice of land lease agreements between land lessees and lessors. In fact, we have generally found that when the transfer of land use rights involves a large amount of economic interest, farmers are more likely to choose a written land lease agreement, otherwise they choose an oral agreement.

The remainder of this paper is organized as follows. Section 2 develops the research hypotheses and theoretical analysis, Section 3 introduces data and methodology, Section 4 presents empirical results and discussion, and Section 5 comprises conclusions and policy implications.

## 2. Background and Conceptual Framework

First, we briefly introduce the general situation of China's land market. China's land market is a dual urban−rural structure, which, in short, is that the ownership of land in the city is owned by the state while the ownership of land in the countryside is owned by the rural collective. That is, China's rural land market and urban land market are actually fragmented [57]. National laws have deprived the reasonable transfer of rural land use rights, which cuts off not only the urban and rural land market in terms of spatial structure but also the value [58]. In the dual segmentation of the land market, the free transfer of land between urban and rural areas is seriously hindered, resulting in an inefficient and unfair allocation of land resources. Farmers' rights to share the income from land appreciation are increasingly restricted, significantly affecting the urban−rural income gap [59]. Therefore, we may conclude that urban land transaction is market-oriented, while rural land transaction is partial property right (use right) market-oriented.

We turn to introducing the development of China's rural land market. As early as 1984, the State Document No. 4 gave rural households the right to transfer farmland. According to the document, rural households can transfer farmland with the permission of the village heads [60]. However, due to factors such as insecurity of land rights, land transactions were very limited before the 1990s [61,62]. In the late 1990s, China's rural land market was growing rapidly, and the proportion of farmland rented in by rural households increased from 1−2% in 1988 [63] to 9.4% in 2000 [64], and 13.5% in 2001−2004 [6]. Wang et al. (2015) found that the proportion of households renting (out) rural land rose from 17% (12%) in 2000 to 27% (19%) in 2008 [1].

To maintain and further improve productivity growth, the Chinese government has gradually introduced a series of land regulations and policies to speed up the transfer of rural land use rights [4,8]. After the second round of land contracting expired, it was required to extend the contract period for another 30 years based on the first round of 15 years in the late 1990s. The Land Management Law of 1998 made it clear that the term of a land lease agreement was 30 years. The employer and the contractor were required to conclude a written land lease agreement to define the rights and obligations

of both parties [6]. The Rural Land Contracting Law of 2003 proposed a series of measures aimed at improving the transferability of rural land use rights. For example, in Ch. 2, § 2, article 19 of this law, the employer should sign a written land lease agreement with the contractor. In 2016, the state council of China issued Opinions on Improving the Method of Dividing Rural Land Ownership Contract Rights and Management Rights. Specific provisions were made on upholding the fundamental status of collective ownership of land, strictly protecting farmers' contracting rights, speeding up the release of land management rights, and improving the relationship between the three rights (i.e., the ownership, contracting and management rights of farmland) [8].

The development of the rural land market has several advantages. First, it provides an efficient allocation of farmland [64,65]. Second, as the land market moves to more efficient producers, rural labor is liberated to participate in non-agricultural employment [1]. Third, the land market facilitates the consolidation of highly fragmented operational land, thereby reducing production costs [61]. To achieve these benefits, China's policies in recent years have encouraged farmers to transfer their rural land use rights. Farmers should strive to speed up the transfer of land use rights to expand the land size, improve agricultural efficiency, and generate higher labor productivity [8].

*2.1. Market-Oriented Economy*

Under the condition of the market economy, both contracting parties hope to maximize the utility of the transfer of land use rights in the process of farmland transfer. This is especially true when the land lease agreement involves greater economic interests. The transfer of land use rights is one of the most important leasing behaviors among rural households. Land lessors who due to some reason are unable to engage in farming their land, rent out their land to get stable rental incomes or to avoid land abandonment; while lessees could rent the land to meet their farming needs, such as for food supplies or expanding farm size. However, from the occurrence of the transfer of land use rights to its termination, it may take months, years, or even decades before the benefits are obtained, during which there will be certain risks and uncertainties.

Therefore, taking into account the risks and uncertainties of any economic transaction, both contracting parties must choose the terms of the land lease agreement to minimize or to avoid the risks and uncertainties as much as possible. Thus, we can propose an economic behavior. Hypothesis 1 (H1): if the transfer of rural land use rights is involved in a large economic interest, both contracting parties are more likely to sign a written land lease agreement.

What determines the economic interest of land lease agreement?

Firstly, it is natural and rational for ones to consider land transfer size as the most important factor. Large scale transfer of land use rights is definitely a large economic interest. Therefore, we can propose another economic behavior. Hypothesis 1a (H1a): if it is a larger scale transfer of land use rights and possibly involves a large economic benefit, both contracting parties are more inclined to sign a written land lease agreement to avoid economic risk and uncertainty.

Secondly, the period of the transfer of land use rights can be the main source of risk and uncertainty. Generally, a longer period of the transfer of land use rights will more likely bring a larger economic risk and uncertainty for both contracting parties, and vice versa. Therefore, farmers always try to seek some kind of contractual arrangement to minimize or avoid risks and uncertainties. Specifically, the long-term land lease agreements can reduce the transferring cost of tenant assets attached to the land and the expected costs of disputes and the transfer of tenant property rights under incomplete information. Therefore, we propose economic behavior Hypothesis 1b (H1b): If it is a long-term land lease agreement and therefore involved a larger economic risk and uncertainty, both contracting parties are more likely to sign a written land lease agreement to avoid economic loss. This is particularly true for land lessees if they want to operate the transferred land for a long period of time and reduce investment risk.

Thirdly, transfer fee is normally negotiated by both contracting parties based on the land size and the transfer of land use rights period. Choosing the longer contractual period land lease agreements

means giving up some of the cost advantages of shorter period land lease agreements. A relatively short period contract can reduce the cost of enforcing contract terms and renegotiating them when the assets on the land owned by tenants run out in a short time, or when the landowner provides all the "permanent" assets [66]. Then, we propose an economic behavior. Hypothesis 1c (H1c): if the transfer of land use rights fee is higher and the land lease agreement is likely involved in larger economic interests, both contracting parties are, therefore, more likely to sign a written land lease agreement in order to avoid economic loss. This is particularly true for land lessors if they want to avoid large rental income losses.

## 2.2. Transitional Economy

Choosing any specific contractual clauses comes at costs and therefore, rational traders need to trade off the costs and benefits when signing contracts and seek the most appropriate clauses. From an institutional cost perspective, the transaction costs include information searching costs, bargaining and negotiation costs, and supervisory costs [67]. Therefore, whether the contracting parties choose to sign a written land lease agreement or not is mainly dependent upon the economic benefits and risk degree of the transfer of land use rights. This is particularly true in a transitional economy, like the one of China.

Therefore, the traditional acquaintance society in rural areas is still playing a role in rural economic activities [8] though market-oriented economy and agricultural modernization have developed for more than four decades. In studying Chinese rural social structure, Fei used "Chaxugeju" to depict the characteristics of "localism" and "acquaintance society" of Chinese society, that is, Chinese social structure itself is different from the western pattern. The Chinese pattern is not that of clear bundles wood, but a rolling motion, as if a stone had been dropped on the surface of the water. Every man is the center of the circle of his social influence [56].

In such a mode, blood tie or kinship becomes the most intimate and stable social relationship, and individuals establish their own social network based on their affinity and distance. Huang (1987) classified these social relationship networks into different associated relations: the expressive tie, the instrumental tie, and the mixed tie [68]. Individuals use different rules of interaction depending on their social relationships under the differential mode of association, as it is in the rural land rental market [8]. There are also different levels of trust in different associations and connections or ties, which are generally divided into two categories, namely, the special trust of the "insider", based on kinship community, and the "common trust", based on common belief and kinship relationship [69]. Trust is attached to social relationships, presenting a reticular pattern [70], and the interpersonal relationship model of affinity and trust eventually forms the differential trust mode of association and the individual's resource acquisition model is based on their position in the relationship network [71]. The differences in trust intensity caused by affinity and alienation affect the recognition of reciprocity and mutual obligations in economic behaviors [72], and make economic behaviors actually become one that is embedded in social networks [73].

It is not unique to China that the transfer of rural land occurs more among acquaintances such as friends and relatives [74]. However, the uniqueness of the differential mode of association of Chinese society, especially the rural society, determines that China's rural land rental market is a special relational human market mixing with geography, kinship, and human relationship [8]. Therefore, according to the above analyses, we propose a social behavior hypothesis 2 (H2): if both contracting parties have a closer social relation, given other conditions are unchanged, they are more likely to have an oral land lease agreement. We develop a conception diagram to more clearly illustrate the construction and fragmentation of the hypotheses in this study (See Figure 1)

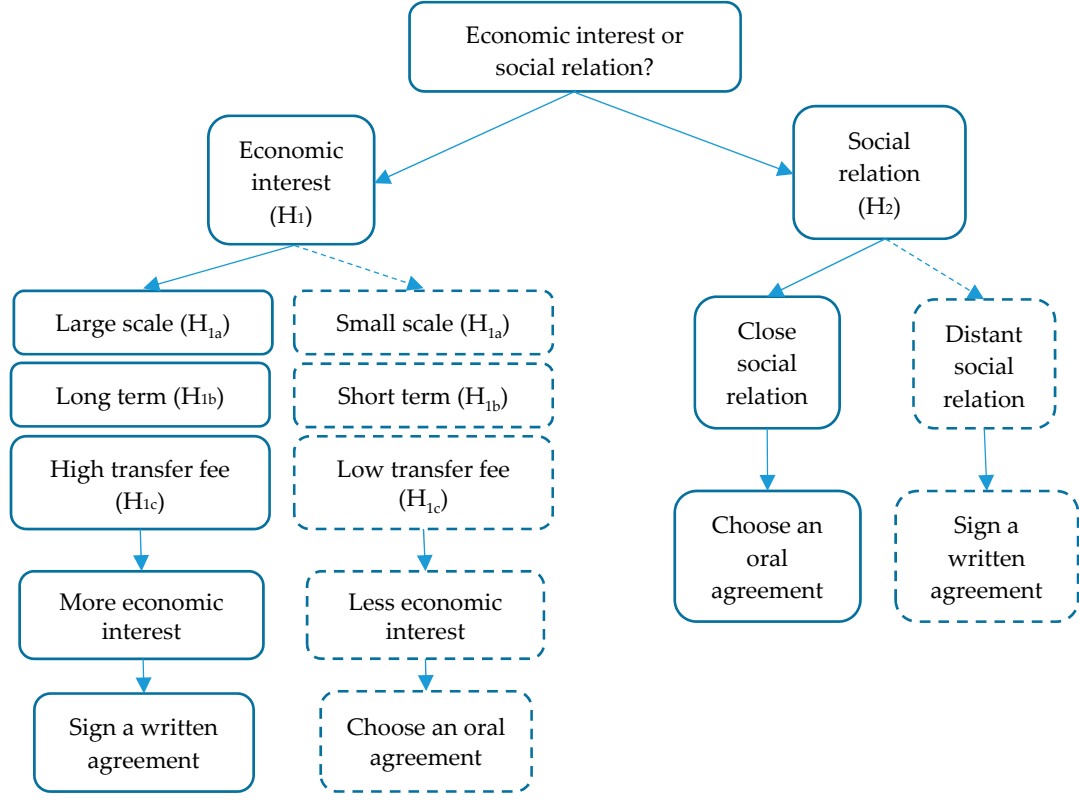

**Figure 1.** Conceptual framework.

## 3. Methods and Data

### 3.1. Data Source

Henan province is located on the middle and lower reaches of the Yellow River and in the south of the North China plain, comprising 21 prefecture-level cities, 87 counties, and 1821 towns. By the end of 2017, the registered population of Henan province was about 95.59 million, of which around 47.64 million engage in agriculture. At the same time, the GDP of Henan had achieved USD 665.16 (1 dollar ≈ 6.75 yuan), ranking the fifth after Guangdong, Jiangsu, Shandong, and Zhejiang provinces.

Henan is an important core area of grain production in China. In 2016, the area utilized for grain cultivation was 10,286.15 million hectares, and the grain output was 59.466 million tons, ranking second in China. Moreover, the Chinese central plains farming culture is centered on Henan province. Henan province rural has the characteristics of traditional Chinese rural society. Henan is the most populous and important economic province in China. In 2016, its GDP ranked fifth among the 31 provinces in China and the first among the central and western provinces.

We conducted a large field survey of rural households in the Henan province of China in August 2014. We applied a careful sampling strategy to capture farmer's rural land use rights transfer. First, twelve out of 21 prefecture-level cities were randomly selected referring to their economic development level and geographic locations. Two counties were randomly selected within each selected prefecture-level city. Second, three towns from each selected county were selected randomly. Third, three villages were selected randomly within each selected town. Finally, ten households from each selected village were sampled randomly for a face-to-face interview. A 2160 rural household sample was used in this study from 216 villages in 72 towns, from 24 counties in 21 prefecture-level cities (10 households × 3 villages × 3 towns × 2 counties × 12 prefecture-level cities in Henan province). In this study, 878 out of the 2160 sample rural households were available to be used to test the choice behavior of land use right transfer contract in major wheat producing province in China.

*3.2. Model Specification*

We specify a binary logit model to test the hypotheses in this study and define the choice behavior of contract type as a function of contracting partners, three contractual clauses, land characteristics, household head and family characteristics, respectively. Therefore, we have primary model as follows:

$$CT_i = \beta_{T0} + \beta_{T1}X_i + \beta_{T2}TP_i + \beta_{T3}TS_i + \beta_{T4}RF_i + \sum_{i=1}\beta_{Tki}Z_{ki} + \varepsilon_{Ti} \tag{1}$$

where $CT_i$ is a dichotomous variable denoting contractual type (written or oral land lease agreements). $X_i$ is a set of the key independent variables denoting different contracting partners, namely relatives or fellow villagers, non-local rural farmers, village collective or government, and cooperatives or companies. It indicates the rural social relations between both contracting partners. $TP_i$ is the transfer period, $TS_i$ is the transfer scale, and $RF_i$ is the land rental fee. $\varepsilon_{Ti}$ is the independent and identically random disturbance term.

This paper also introduces a series of control variables $Z_{ki}$. First, land characteristics, such as land fragmentation degree (number of land plots owned by a single household), land ownership confirmation, and land subsidy. Second, household characteristics, including the age of the household head and his/her education level, technical training experience, and farming working time. Third, family characteristics, including the percentage of crop plantation income in the total family income and the number of family laborers. Fourth, regional dummy variables (west, north and south). Table 1 presents variable definitions and summary statistics.

Moreover, we consider the interaction of the transfer period, rental fee and transfer scale may have an impact on farmers' choice of land lease agreements. Thus, we treat the interaction between transfer period and transfer scale, transfer scale and rental fee, and transfer period and rental fee as newly generated comprehensive variables, respectively, and include three multiplicative interaction terms in the model (2).

$$CT_i = \alpha_{T0} + \alpha_{T1}X_i + \alpha_{T2}TP_i + \alpha_{T3}TS_i + \alpha_{T4}RF_i + \alpha_{T5}TP \times TS_i + \alpha_{T6}RF \times TS_i$$
$$+ \alpha_{T7}TP \times RF_i + \sum_{i=1}\alpha_{Tki}Z_{ki} + \varepsilon_{Ti} \tag{2}$$

where $TP \times TS_i$ is the interaction term between transfer period and transfer scale, $RF \times TS_i$ is the interaction term between rental fee and transfer scale, and $TP \times RF_i$ is the interaction term between transfer period and rental fee, respectively. The other variables have the same meaning as in Equation (1).

Besides using transfer scale as one of the indicators to measure economic benefits, in this study, we also employ the proportion of the transferred land to the total cultivated land to measure the importance of transferred land to the lessors. Thus, we set Equation (3) as follows:

$$CT_i = \delta_{T0} + \delta_{T1}X_i + \delta_{T2}TP_i + \delta_{T3}TSP_i + \delta_{T4}RF_i + \sum_{i=1}\delta_{Tki}Z_{ki} + \varepsilon_{Ti} \tag{3}$$

where $TSP_i$ denotes the proportion of the transferred land. For land lessors, $TSP_i$ can be calculated as $TSP_i$ = rented out land areas divided by total cultivated land areas. For land lessees, $TSP_i$ = (rented minus rented out) land areas divided by total cultivated land.

Considering their importance, we also incorporate interaction terms of the proportion of the transferred land with the transfer period and rental fee, respectively, into our models. Then, we have Equation (4) as follows:

$$CT_i = \gamma_{T0} + \alpha\gamma_{T1}X_i + \gamma_{T2}TP_i + \gamma_{T3}TSP_i + \gamma_{T4}RF_i + \gamma_{T5}TP \times TSP_i + \gamma_{T6}RF \times TSP_i$$
$$+ \gamma_{T7}TP \times RF_i + \sum_{i=1}\gamma_{Tki}Z_{ki} + \varepsilon_{Ti} \tag{4}$$

where $TP \times TSP_i$ denotes the interaction term of transfer period and the proportion of the transfer scale to the total cultivated land. $RF \times TSP_i$ is the interaction term of rental fee and the proportion of the transferred land. The other variables have the same meaning as in Equation (1).

**Table 1.** Variable definitions and statistical description between land lessors and lessees.

| Variables | Definitions | Land Lessors | | Land Lessees | |
|---|---|---|---|---|---|
| | | Mean | Std. Dev. | Mean | Std. Dev. |
| (1) Dependent variable Contractual type (*CT*) | 1 = written, 0 = oral | 0.70 | 0.46 | 0.23 | 0.42 |
| (2) Independent variables Contracting partners: | | | | | |
| Relatives or fellow villagers (*Relative_vill*) | 1 = yes, 0 = otherwise | 0.48 | 0.50 | 0.93 | 0.25 |
| Non-local rural farmers (*Non_local*) | 1 = yes, 0 = otherwise | 0.23 | 0.42 | 0.03 | 0.16 |
| Village or government (*Village_gov*) | 1 = yes, 0 = otherwise | 0.09 | 0.28 | 0.03 | 0.17 |
| Cooperatives or companies (*Coop_com*) (baseline) | 1 = yes, 0 = otherwise | 0.20 | 0.40 | 0.01 | 0.11 |
| Contract clauses: | | | | | |
| Transfer period (*TP*) | years | 10.88 | 9.28 | 4.09 | 5.74 |
| Rental fee (*RF*) | ln (yuan/mu) | 6.41 | 1.29 | 5.34 | 1.98 |
| Transfer scale (*TS*) | mu | 5.22 | 4.69 | 32.46 | 58.44 |
| The proportion of the transfer scale to the total cultivated land (*TSP*) | % | 0.78 | 0.27 | 0.61 | 0.25 |
| (3) Control variables Land characteristics: | | | | | |
| Land fragmentation (*Land_frag*) | No. of plots | 2.76 | 1.68 | 2.84 | 2.25 |
| Land ownership confirmation (*Land_conf*) | 1 = yes, 0 = no | 0.30 | 0.46 | 0.31 | 0.46 |
| Land subsidy (*Land_sub*) | ln (yuan/mu) | 4.63 | 0.59 | 4.52 | 0.91 |
| Household characteristics: | | | | | |
| Age (*Age*) | years | 49.16 | 9.36 | 49.58 | 8.13 |
| Education level (*Edu*) | years | 7.94 | 2.98 | 8.24 | 2.90 |
| Technical training experience (*Tech_train*) | 1 = yes, 0 = no | 0.44 | 0.50 | 0.57 | 0.50 |
| Agricultural working time (*Agr_time*) | days per year | 41.08 | 73.18 | 76.41 | 69.60 |
| Family characteristics: | | | | | |
| The proportion of the crop plantation income to the total household income (*Cropp*) | % | 56.40 | 48.46 | 87.63 | 28.83 |
| No. of family laborers (*Laborers*) | person (s) | 3.30 | 1.50 | 3.52 | 1.43 |
| Region: | | | | | |
| *Eastern* (baseline) | 1 = yes, 0 = otherwise | 0.23 | 0.42 | 0.31 | 0.46 |
| *Western* | 1 = yes, 0 = otherwise | 0.35 | 0.48 | 0.24 | 0.43 |
| *Southern* | 1 = yes, 0 = otherwise | 0.07 | 0.26 | 0.24 | 0.43 |
| *Northern* | 1 = yes, 0 = otherwise | 0.34 | 0.48 | 0.21 | 0.41 |

## 4. Results and Analyses

### 4.1. Descriptive Sstatistics

Table 2 displays the sample distribution and the characteristics of both land lessees and lessors. Of the 878 rural households that transfer their land, 492 (56.04%) are land lessees, while 386 (43.96%) are land lessors. With regard to contract type, 76.83% of the land lessees choose oral agreements, while 69.95% of the land lessors choose the written land lease agreements.

**Table 2.** Sample structure and characteristics of the land lease agreements and contracting partners.

| Variables | Land Lessors | | Land Lessees | |
|---|---|---|---|---|
| | Number | % | Number | % |
| (1) Land lease agreement characteristics | | | | |
| Contractual type: | | | | |
| Written agreement | 270 | 69.95 | 114 | 23.17 |
| Oral agreement | 116 | 30.05 | 378 | 76.83 |
| Transfer period: | | | | |
| Short-term (≤10 years) | 237 | 61.40 | 447 | 90.85 |
| Medium-term (11−20 years) | 96 | 24.87 | 35 | 7.11 |
| Long-term (>20 years) | 53 | 13.73 | 10 | 2.03 |
| Rental fee: | | | | |
| Low (≤500 yuan per mu) | 74 | 19.17 | 337 | 68.50 |
| Medium (500−1000 yuan per mu) | 235 | 60.88 | 139 | 28.25 |
| High (>1000 yuan per mu) | 77 | 19.95 | 16 | 3.25 |
| Transfer scale: | | | | |
| Small (≤5 mu) | 265 | 68.65 | 177 | 35.98 |
| Medium (5−10 mu) | 88 | 22.80 | 91 | 18.50 |
| Large (>10 mu) | 33 | 8.55 | 224 | 45.53 |
| (2) Contracting partners | | | | |
| Relatives or fellow villagers | 186 | 48.19 | 458 | 93.09 |
| Non-local rural farmers | 90 | 23.32 | 13 | 2.64 |
| Village collective or government | 34 | 8.81 | 15 | 3.05 |
| Cooperatives or companies | 76 | 19.69 | 6 | 1.22 |
| (3) Region | | | | |
| Eastern | 89 | 23.06 | 152 | 30.89 |
| Western | 136 | 35.23 | 119 | 24.19 |
| Southern | 28 | 7.25 | 118 | 23.98 |
| Northern | 133 | 34.46 | 103 | 20.93 |
| Observation | 386 | 100 | 492 | 100 |

Notes: In this study, the whole Henan province is geographically disaggregated into four regions: east, west, south and north. The eastern region includes Zhoukou city (sampling Dancheng and Xiangcheng counties), Shangqiu city (sampling Yongcheng city and Zhecheng county), and Xuchang city (sampling Xuchang county and Changge city). The western region includes Sanmenxia city (sampling Lingbao city and Lushi county), Luoyang city (sampling Luoning and Yiyang counties), and Pingdingshan city (sampling Wugang city and Baofeng county). The southern region includes Zhumadian city (sampling Zhengyang and Queshan counties), Nanyang city (sampling Dengzhou city and Xixia county), and Xinyang city (sampling Xin and Gushi counties). The northern region includes Jiaozuo city (sampling Wenxian county and Mengzhou city), Xinxiang city (sampling Yuanyang county and Weihui city), and Anyang city (sampling Huaxian and Neihuang counties.

For contracting partners, most farmers rent the land from their relatives or fellow villagers (households in the same village), accounting for 93.09%, while 48.19% of the lessors rent out the land to their relatives or fellow villagers, followed by non-local rural farmers, cooperatives or companies, and village collectives or governments, accounting for 23.32%, 19.69% and 8.81%, respectively.

Turning to the transfer period, the majority of the land lessees opt for short-term (10 years and below) land lease agreements (90.85%), while 7.11% households choose medium-term (10–20 years) land lease agreements, and only 2.03% households choose long-term (more than 20 years) land

lease agreements. It seems that 61.40%, 24.87%, and 13.73% of the land lessors prefer short-term, medium-term, and long-term land lease agreements, respectively.

Regarding rental fee, 68.50% of the land lessees pay the low rental fee (500 yuan and below per mu), followed by the medium rental fee (500–1000 yuan per mu) (28.25%), and the high rental fee (more than 1000 yuan per mu) (3.25%). Among land lessors, 60.88% report that they receive medium rental income, while 19.95% and 19.17% receive high rental income and low rental income, respectively.

Looking at transfer scale, 45.53% of land lessees have a large transfer scale (more than 10 mu), followed by the small transfer scale (5 mu and below) and the medium transfer scale (5−10 mu), accounting for 35.98% and 18.50%, respectively. It seems different from land lessees, of which 68.65%, 22.80%, and 8.55% have the small transfer scale, medium transfer scale and large transfer scale, respectively. In fact, individual household cannot rent out more than 10 mu land, on average (10 mu per household), while individual households can rent as much land as they want if the lands are available. This will lead to a trend of rising concentration of land to the land lessees.

We also observe that the highest proportion of land lessees is 30.89% in the east of Henan province, while the lowest proportion is 20.93% in the north of Henan province. The highest proportion of land lessors is 35.23% in the west of Henan province, while the lowest proportion is 7.25% in the north of Henan province.

### 4.2. Choice of Contractual Type for Lessors

Table 3 reports the estimated results of the choice of land lease agreement for land lessors. Looking at the results of the contractual clauses, we find that transfer period, transfer scale, and rental fee positively affect the choice of a written land lease agreement (Model 1). It suggests that a longer transfer period, larger transfer scale, and higher rental fee involve a large economic interest, which make land lessors more likely to choose a written land lease agreement. The results show strong evidence to verify H1a, H1b, and H1c. In fact, there were 98 cases of large size transfers of rural land use rights (over 40 mu) in our field survey of the transfer of rural land use rights, and we found that in 65 cases written land lease agreements were signed, accounting for 66% of total large scale land use rights transfers. In contrast, there were 426 cases of small size land lessees (below 20 mu), however, we found that in 352 cases written land lease agreements were not signed, accounting for 82% of total small scale land use right transfers. Our field survey also showed that among 10–20-year period cases, 107 land lessors signed a written land lease agreement but only two lessors did not sign. Similarly, among over 20-year period cases, 35 land lessors signed a written land lease agreement but only four lessors did not sign. Obviously, with the development of China's rural society and economy, the market economy has a growing influence on rural society. The demand for economic returns and risk reduction will prompt more farmers to choose a written land lease agreement. We have also observed that the rental fee has the largest influence on choice of written land lease agreements for land lessors, followed by the transfer scale and rental period. In addition, the coefficient of rental fee (See RF variable, Model 2) consistently shows a positive and significant impact on the choice of written land lease agreements for land lessors. This indicates that land lessors are more careful of economic interests. Therefore, they are more likely to sign a written land lease agreement to secure their land rental income. For interaction terms, however, these coefficients between transfer period−transfer scale and period−rental fee are both not significant.

**Table 3.** The estimated results for the choice of the written land lease agreement for land lessors.

| Variables | Model 1 | | Model 2 | | Model 3 | | Model 4 | |
|---|---|---|---|---|---|---|---|---|
| | Coefficient | Odds Ratio | Coefficient | Odds Ratio | Coefficient | Odds Ratio | Coefficient | Odds Ratio |
| **Contracting partners:** | | | | | | | | |
| *Relative_vill* | −0.743 * (0.42) | 0.476 | −0.616 (0.42) | 0.540 | −0.879 ** (0.41) | 0.415 | −0.698 * (0.41) | 0.497 |
| *Non_local* | 0.517 (0.46) | 1.676 | 0.564 (0.46) | 1.759 | 0.380 (0.45) | 1.462 | 0.449 (0.46) | 1.566 |
| *Village_gov* | −0.402 (0.59) | 0.669 | −0.384 (0.59) | 0.681 | −0.386 (0.57) | 0.680 | −0.440 (0.57) | 0.644 |
| **Contract clauses:** | | | | | | | | |
| *TP* | 0.042 ** (0.02) | 1.043 | 0.219 (0.14) | 1.245 | 0.045*** (0.02) | 1.046 | 0.605*** (0.19) | 1.830 |
| *TS* | 0.133 ** (0.06) | 1.142 | −0.016 (0.33) | 0.984 | – | – | – | – |
| *TSP* | – | – | – | – | −0.406 (0.56) | 0.666 | 24.83 *** (7.70) | 0.000 |
| *RF* | 0.796 *** (0.28) | 2.218 | 1.397 *** (0.52) | 4.044 | 0.810 *** (0.30) | 2.248 | −0.671 (0.62) | 2.248 |
| **Interaction terms:** | | | | | | | | |
| *TP×TS* | – | – | 0.006 (0.01) | 1.006 | – | – | – | – |
| *RF×TS* | – | – | 0.012 (0.01) | 1.012 | – | – | – | – |
| *TP×RF* | – | – | −0.031 (0.02) | 0.969 | – | – | −0.086 *** (0.03) | 0.918 |
| *TP×TSP* | – | – | – | – | – | – | 0.001 (0.06) | 1.001 |
| *RF×TSP* | – | – | – | – | – | – | 3.697 *** (1.16) | 40.313 |
| **Control variables** | | | | | | | | |
| **Land characteristics:** | | | | | | | | |
| *Land_frag* | −0.063 (0.09) | 0.939 | −0.050 (0.09) | 0.951 | −0.062 (0.08) | 0.974 | −0.034 (0.09) | 0.966 |
| *Land_conf* | 0.228 (0.30) | 1.256 | 0.233 (0.30) | 1.263 | 0.085 (0.30) | 1.088 | 0.107 (0.31) | 1.113 |
| *Land_sub* | 0.128 (0.26) | 1.137 | 0.147 (0.26) | 1.158 | 0.204 (0.26) | 1.226 | 0.222 (0.27) | 1.249 |
| **Household characteristics:** | | | | | | | | |
| *Age* | 0.016 (0.02) | 1.016 | 0.017 (0.02) | 1.017 | 0.017 (0.01) | 1.018 | 0.016 (0.02) | 1.015 |
| *Edu* | −0.022 (0.05) | 0.978 | −0.030 (0.05) | 0.971 | −0.017 (0.05) | 0.984 | −0.016 (0.05) | 0.984 |
| *Tech_train* | 0.911 *** (0.29) | 2.488 | 0.893 *** (0.30) | 2.441 | 0.975 *** (0.29) | 2.652 | 0.930 *** (0.30) | 2.535 |
| *Agr_time* | 0.000 (0.00) | 1.000 | −0.000 (0.00) | 1.000 | 0.000 (0.00) | 1.000 | −0.000 (0.00) | 1.000 |
| **Family characteristics:** | | | | | | | | |
| *Cropp* | −0.001 (0.00) | 0.999 | −0.001 (0.00) | 0.999 | −0.003 (0.00) | 0.996 | −0.003 (0.00) | 0.999 |
| *Laborers* | 0.259 ** (0.11) | 1.295 | 0.249 ** (0.11) | 1.283 | 0.249 ** (0.11) | 1.283 | 0.248 ** (0.11) | 1.281 |
| **Region:** | | | | | | | | |
| *Western* | −0.117 (0.39) | 0.890 | −0.007 (0.40) | 0.993 | −0.175 (0.39) | 0.839 | −0.031 (0.40) | 0.969 |
| *Southern* | −0.531 (0.39) | 0.588 | −0.310 (0.41) | 0.733 | −0.755 * (0.40) | 0.470 | −0.480 (0.41) | 0.619 |
| *Northern* | −1.096 * (0.59) | 0.334 | −0.730 (0.63) | 0.482 | −1.256 ** (0.61) | 0.285 | −0.722 (0.68) | 0.486 |
| Constant | −6.928 *** (2.43) | 0.000 | −10.807 *** (3.80) | 0.000 | −6.338 ** (2.60) | 0.002 | 3.303 (4.15) | 27.207 |
| LR chi$^2$ | 121.96 | | 126.17 | | 115.65 | | 130.14 | |
| Pseudo R$^2$ | 0.258 | | 0.267 | | 0.245 | | 0.276 | |
| Log likelihood | −174.985 | | −172.877 | | −178.136 | | −170.894 | |
| Observations | 386 | | 386 | | 386 | | 386 | |

*Notes*: Standard errors in parentheses. *** $p < 0.01$, ** $p < 0.05$, * $p < 0.1$.

For contractual clauses in Model 3 and Model 4. Transfer period positively affects lessors' choice of the land lease agreement, while rental fee has a positive influence on the choice of land lease agreement for land lessors. Interestingly, we have found that the proportion of the transferred land has a negative influence on the choice of land lease agreement for land lessors. These results show that a higher proportion of transferred land is less likely to make land lessors choose a written contract. When they rent out a large part of their land, land lessors will lose the control of their land temporally. Therefore, this finding probably indicates that an oral agreement may make it easier for land lessors to reclaim their transferred land whenever they need. In other words, land may be still a basic livelihood source for rural households.

With regard to interaction terms, the effect between transfer period and rental fee is negatively significant. This result shows that higher rental fee and longer transfer period will have land lessors less likely choose a written land lease agreement. Obtaining land rental income is one of the most important purposes for land lessors. However, given current agricultural supporting policy, a longer transfer period may bring about a greater uncertainty and risk to land lessors, and therefore, to get more future land income, land lessors may no longer have a strong expectation to sign a written land lease agreement even it can reduce short-term uncertainty and risk. Furthermore, the interaction effect between the rental fee and the proportion of transferred land is positively significant. As discussed above, due to the flexible requirements of land management a higher proportion of transferred land is more likely, making land lessors choose an oral agreement of land rent. On the other hand, high land rental income can compensate for the risk of transferred land for land lessors. Thus, the flexible land management becomes particularly important for land lessors. Therefore, a higher rental fee and larger proportion of transferred land will make land lessors more likely to choose an oral agreement of land rent.

We have found that social relations between land lessors and their relatives or fellow villagers have a negative effect on their choice of the land lease agreement (Model 1, Model 3 and Model 4). The results show that the probability of selecting an oral land lease agreement between land lessors and their relatives or fellow villagers is larger than that between them and cooperatives or companies. This finding supports H2 to some extent. This finding confirms that farmers still prefer to have an oral land lease agreement with their acquaintances when renting out their land [6,18], indicating that the paces of marketization and standardization of the transfer of land use rights are still slow in China [8,22]. It is consistent with the theory of Fei [56]. In fact, our survey data show that the percentages of oral contract decreases from the closest to the farthest social relation. In detail, the percentages of land lessees who chose an oral land lease agreement decreased, with values of 100%, 80.5%, 46.2%, 16.7% and 13.3% for relatives, villagers, outer village, cooperative or company and collective or government, respectively. However, the percentages of lessees who chose a written land lease agreement increased, with values of 0%, 19.5%, 53.8%, 83.3% and 86.7% for relatives, villagers, outer village, cooperative or company and collective or government, respectively. In addition, we have found that social relations between land lessors and other contractual parties have no significant influence on the choice of the land lease agreement.

It seems strange to find that the number of family laborers has a positive effect on the choice of the written land lease agreement for land lessors. Essentially, this result indicates that land lessors of more family laborers are likely to choose a written land lease agreement for their rented out land. However, we have found that most family laborers are engaging in non-farm work for these land lessors, making them more likely sign a written contract so that they can reduce the future transfer of land use rights transaction cost and obtain a stable rental income. In fact, survey result shows that average 3.4 family laborers are engaging in off-farm work for these land lessors.

In addition, we have observed that technical training experience and farming working time have a positive impact on the choice of the written land lease agreements for land lessors. However, we have not found that land fragmentation has a significant impact on the choice of their land lease agreements.

*4.3. Choice of Contractual Type for Lessees*

Table 4 displays the estimated results of the choice of land lease agreements for land lessees. In terms of contract clauses, the same result can be found as above for lessees. As shown in Model 1 and Model 2 (Table 4), the transfer period, transfer scale, and rental fee have a positive impact on the choice of land lease agreements. These results indicate that the longer transfer period, the larger transfer scale, and the higher rental fee are all more likely to make land lessees choose a written land lease agreement. This finding verifies $H_{1a}$, $H_{1b}$ and $H_{1c}$ in this study. Moreover, we have found that the rental fee has the biggest influence on the choice of the written land lease agreements, followed by the transfer period and transfer scale. In addition, the coefficient of the transfer period (See *CP* variable in Table 4) consistently shows a positive and significant impact on the choice of the written land lease agreements for land lessees. This suggests that land lessees are more careful of transfer period, which allows them to have sufficient land operating time to ensure consistent and stable agricultural production.

Observing the results of Model 3 and Model 4 (Table 4), we can find that transfer period, the proportion of transferred land, and rental fee have a positive impact on the choice of the land lease agreements. Especially, we find that a higher proportion of transferred land is more likely to make land lessees choose a written land lease agreement. This finding implies that in a sense, a higher proportion of transferred land will involve larger economic interests to land lessees. Therefore, to guarantee a long-term land use right for large-scale operations, land lessees are more inclined to sign a written land lease agreement. In addition, interaction terms do not significantly affect the choice of written land lease agreement for land lessees.

The results show that the social relations between land lessees and relatives or fellow villagers in the four models have a negative significant influence on their selection of the land lease agreements. This result supports $H_2$. That is, to land lessees, the possibility of choosing an oral agreement with their relatives or villagers is higher than that with cooperatives or companies. These findings are consistent with that as found for land lessors above.

It seems strange to find that the number of family laborers has a negative effect on the choice of the written land lease agreement for land lessees. This result indicates that land lessees made up of more family laborers are likely to choose an oral agreement for their rented land. The reasons are mixed. Possible explanations may be: (i) they are not interested in the rented land or plots or (ii) most of the family laborers are engaging in non-farm work. In contrast, land lessees made up of fewer family laborers are likely to choose a written land lease agreement. This may be because if these households with a smaller non-agricultural population want to work in agriculture they can obtain agricultural benefits by renting more land. Therefore, land tenants want to sign formal written land lease agreements to reduce the risk of renting more land. In addition, more agricultural machinery inputs may let less laborer land lessees sign a written land lease agreement so that they can obtain a stable and long-term land use right.

In addition, we have observed that technical training experience and farming working time have a positive impact on the choice of the written land lease agreements for land lessees. However, we have not found that land fragmentation has a significant impact on the choice of land lease agreements for land lessees.

**Table 4.** The estimated results for the choice of the written land lease agreement for land lessees.

| Variables | Model 1 | | Model 2 | | Model 3 | | Model 4 | |
|---|---|---|---|---|---|---|---|---|
| | Coefficient | Odds Ratio | Coefficient | Odds Ratio | Coefficient | Odds Ratio | Coefficient | Odds Ratio |
| **Contracting partners:** | | | | | | | | |
| *Relative_vill* | −3.116 ** (1.35) | 0.044 | −2.961 ** (1.34) | 0.052 | −2.704 * (1.41) | 0.067 | −2.814 ** (1.40) | 0.600 |
| *Non_local* | −1.937 (1.50) | 0.144 | −1.750 (1.49) | 0.174 | −1.409 (1.53) | 0.245 | −1.446 (1.52) | 0.235 |
| *Village_gov* | 0.170 (1.64) | 1.185 | 0.348 (1.64) | 1.416 | 0.514 (1.69) | 1.673 | 0.468 (1.66) | 1.596 |
| **Contract clauses:** | | | | | | | | |
| *TP* | 0.136 *** (0.02) | 1.146 | 0.118 ** (0.05) | 1.126 | 0.158 *** (0.02) | 1.171 | 0.105 (0.07) | 1.111 |
| *TS* | 0.014 *** (0.00) | 1.014 | −0.028 (0.03) | 0.972 | – | – | – | – |
| *TSP* | – | – | – | – | 2.215 *** (0.65) | 9.164 | −0.380 (3.28) | 0.684 |
| *RF* | 0.266 ** (0.12) | 1.304 | 0.157 (0.15) | 1.171 | 0.277 ** (0.12) | 1.319 | −0.000 (0.28) | 1.000 |
| **Interaction terms:** | | | | | | | | |
| *TP×TS* | – | – | −0.000 (0.00) | 0.999 | – | – | – | – |
| *RF×TS* | – | – | 0.007 (0.00) | 1.007 | – | – | – | – |
| *TP×RF* | – | – | 0.005 (0.01) | 1.005 | – | – | 0.008 (0.01) | 1.008 |
| *TP×TSP* | – | – | – | – | – | – | 0.022 (0.11) | 1.022 |
| *RF×TSP* | – | – | – | – | – | – | 0.412 (0.51) | 1.510 |
| **Control variables** | | | | | | | | |
| **Land characteristics:** | | | | | | | | |
| *Land_frag* | −0.076 (0.08) | 0.927 | −0.059 (0.08) | 0.943 | −0.016 (0.08) | 0.984 | −0.020 (0.08) | 0.981 |
| *Land_conf* | −0.541 (0.36) | 0.582 | −0.502 (0.36) | 0.606 | −0.336 (0.33) | 0.714 | −0.343 (0.34) | 0.709 |
| *Land_sub* | −0.085 (0.14) | 0.918 | −0.084 (0.15) | 0.920 | −0.070 (0.14) | 0.932 | −0.069 (0.15) | 0.933 |
| **Household characteristics:** | | | | | | | | |
| *Age* | 0.002 (0.02) | 1.002 | −0.000 (0.02) | 0.999 | −0.003 (0.02) | 0.997 | −0.004 (0.02) | 0.996 |
| *Edu* | 0.031 (0.06) | 1.032 | 0.026 (0.06) | 1.027 | 0.040 (0.06) | 1.041 | 0.035 (0.06) | 1.035 |
| *Tech_train* | 0.729 ** (0.33) | 2.074 | 0.783 ** (0.34) | 2.188 | 0.749 ** (0.32) | 2.115 | 0.738 ** (0.32) | 2.092 |
| *Agr_time* | 0.004 ** (0.00) | 1.004 | 0.004 ** (0.00) | 1.004 | 0.005 *** (0.00) | 1.005 | 0.005 ** (0.00) | 1.005 |
| **Family characteristics:** | | | | | | | | |
| *Cropp* | 0.004 (0.01) | 1.004 | 0.003 (0.01) | 1.003 | 0.005 (0.01) | 1.005 | 0.005 (0.01) | 1.005 |
| *Laborers* | −0.259 ** (0.13) | 0.772 | −0.271 ** (0.13) | 0.762 | −0.214 * (0.12) | 0.807 | −0.215 * (0.12) | 0.806 |
| **Region:** | | | | | | | | |
| *Western* | −1.707 *** (0.41) | 0.181 | −1.806 *** (0.42) | 0.164 | −1.528 *** (0.38) | 0.217 | −1.555 *** (0.38) | 0.211 |
| *Southern* | −1.188 *** (0.44) | 0.305 | −1.198 *** (0.44) | 0.302 | −1.586 *** (0.44) | 0.205 | −1.557 *** (0.44) | 0.211 |
| *Northern* | −1.537*** (0.49) | 0.215 | −1.513*** (0.49) | 0.220 | −1.756*** (0.48) | 0.173 | −1.766*** (0.48) | 0.171 |
| Constant | 0.035 (1.95) | 1.036 | 0.684 (2.05) | 1.981 | −1.687 (1.98) | 0.185 | 0.194 (2.67) | 1.214 |
| LR chi2 | 220.14 | | 223.25 | | 199.71 | | 200.80 | |
| Pseudo R2 | 0.413 | | 0.419 | | 0.375 | | 0.377 | |
| Log likelihood | −156.266 | | −154.708 | | −166.482 | | −165.933 | |
| Observations | 492 | | 492 | | 492 | | 492 | |

Notes: Standard errors in parentheses. *** $p < 0.01$, ** $p < 0.05$, * $p < 0.1$.

## 5. Conclusions and Implications

This paper investigated the choice behavior of land lease agreements in the transfer of rural land use rights by testing the role of economic interests and social relations. From the estimated results and descriptive statistics, we provide some new and interesting conclusions and implications as follows:

Firstly, economic interests play a more important role in the choice behavior of land lease agreements in rural China. The estimated results show that transfer period, rental fee, and transfer scale are all positive and significant, indicating economic interests play a key role in choosing the written land lease agreement. This is true for both land lessees and land lessors (recall Tables 3 and 4). In fact, about 67% of total rented out land comes from those households who rent out their half land, about 30% of total rented out land comes from those households who rent out all land, and about 60% of all rented out land has an over six-year transfer period. All these descriptive statistics involved a large number of economic interests. Therefore, these results indicate that economic behavior is the major determinant of the transfer of land use rights in China.

However, rural social relations still affect the decision of farmers choosing a written land lease agreement or oral agreement for their transfer of land use rights, but not by much. Taking account of transaction cost, the farmers may be likely to choose an oral agreement rather than a written land lease agreement when they transfer their land use rights to their acquaintances (*Relative_vill*, Tables 3 and 4). This is true for both land lessees and land lessors. However, does social relation affect the transfer of rural land use rights? In-depth observation of land use rights transfer scale may tell a different story. For example, for 378 land lessees, only 27.7% of the total 26788 mu of land rented is done so without a signed written land lease agreement. For 116 land lessors, only 18.9% of the total 2358 mu of land rented out is done so without signing a written land lease agreement. These results indicate that social relation is a minor determinant of the transfer of land use rights in rural China from the perspective of land use rights transfer scale.

Thirdly, the importance of land to farmers affects the decision of whether farmers choose a written land lease agreement for their transfer of land use rights. The proportion of rented out land negatively affects the choice of the written land lease agreement for land lessors (*TSP*, Model 4 of Table 3), which probably indicates that rented out land is important to them. They are not careful to sign a written land lease agreement so that they may reclaim the land use right whenever they need. In contrast, the proportion of rented in land (*TSP*, Model 3 of Table 4) positively influences the choice of the written land lease agreements for land lessees, which probably means that rented land is important for land lessees. They are more likely to sign a written land lease agreement to obtain a stable land use right.

Finally, to speed up the transfer of land use rights and scale of operation, we have some major policy suggestions: (i) reduce the dependence of land for rural households; (ii) improve agricultural machinery and achieve farming mechanization; (iii) create more non-farm work opportunities for rural labors, comprehensively promote the process of citizenization (i.e., the process in which the agricultural population transfers from the countryside to the cities and gradually becomes urban residents, that is to say, the farmers' *hukou* changes from the countryside to the city) of rural households; (iv) farmers' choice of contracts should be respected so as to better safeguard their interests; and (v) the government should attach importance to the acquaintance society among farmers, and the function of social relations cannot be ignored in the process of rural land transfer.

Our study takes Henan province of China as a case study. It has important enlightenment and reference significance for the relevant studies on China's transfer of rural land use rights. Although our research is limited to Henan province, China, the issues discussed above may even be broadly relevant to developing countries, such as India, Ethiopia, Vietnam, and Bangladesh. These countries are aiming at the transition from an agricultural structure to a more diversified economic structure, but for a variety of reasons have limited the scope of the land rental market. Moreover, the evidence presented here suggests that national laws and regulations require a signed written land lease agreement for the transfer of land, but farmers have their own way. A large study of the impact of relationship network and economic interests on the choice of land lease agreements and factors that may promote or limit

land transferability would, especially in these countries mentioned above, be a promising research approach with direct policy implications.

Future research can improve this study from at least two aspects. For one thing, a well-known problem with rural household surveys on land transactions in China is the fact that lessors with a rural *hukou* who have migrated elsewhere and rented out all their land cannot be found at their rural homes at the survey time, and therefore cannot be interviewed. On the other hand, we haven't considered the interaction effects between kinship and contract terms. Although, Holden and Ghebru (2006) have already examined the interaction effects between kinship and the transfer of land use rights period [49]. Furthermore, Ma et al. (2019) argued that the choice of contract partners may also depend on economic factors like costs related to the physical distance between the partners, whereas transfer period, transfer scale, and rental fee partly depend on social relations in addition to economic interests [54].

**Author Contributions:** Conceptualization, R.L. and H.M.; Validation, R.L. and H.M.; Formal Analysis, R.L., Z.G. & H.M.; Investigation, R.L. and H.M., Resource, R.L. and H.M.; Data Curation, R.L. and Y.N.; Writing—Original Draft Preparation, R.L. and Z.G.; Writing—Review & Editing, R.L., Z.G., Y.N., & H.M.; Project Administration, R.L. and H.M.; Funding Acquisition, H.M. All authors have read and agreed to the published version of the manuscript.

**Funding:** This research was funded by National Natural Science Foundation of China (Grant No. 71403082); Specialized Research Fund for the Jointed Doctoral Program of Higher Education of China (Grant No. 20124105110006); Ministry of Education of Humanities and Social Science Research Project of China (Grant No. 14YCJ790080); Annual Scientific and Technological Innovation of Henan Province Talent Support Program (Grant No. 2017-cxrc-002); Young Backbone Teachers Scheme of Henan Colleges and Universities (Grant No. 2015GGJS-085); Henan Province Philosophy and Social Science Planning Project (Grant No. 2017BJJ033); Henan Provincial Department of Education Humanities and Social Sciences Research Project (Grant No. 2019-ZZJH-327); China Scholarship Council.

**Conflicts of Interest:** The authors declare no conflict of interest.

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
