# Peer review of "Does Social Relation or Economic Interest Affect the Choice Behavior of Land Lease Agreement in China? Evidence from the Largest Wheat−Producing Henan Province"

_sustainability, doi:10.3390/su12104279_

Round 1

Reviewer 1 Report

1. well explained and structured paper. Just feel to have an overview diagram of methodology.

2. Please explain briefly HRS. And what is the evidence that HRS was responsible to improve factor productivity and output growth? Does HRS include division of labour principle, economic principles like marginal productivity and marginal cost? or technical factors like High yielding varieties, intensive cropping patterns, mechanization etc.? or land governance factors like land consolidation, land distribution among landless?

3. please mention the year of the land policy? line 52

4. language not clear. line 335 & 442.

5. Check the consistency of Hypothesis numbers.

6. The results provide strong evidence that economic interest plays a key role in the choice of rental contract. However, the title doesn't include this key word or provide a hint. In fact, the paper involves between socio-economic interactions but the title doesn't give an idea about this.

Reviewer 2 Report

The article discusses the development of the land lease market in China. I appreciate the work of the authors, but I regret to point out that the following issues raise doubts:
- It is difficult to justify the compatibility of the content of the article with the topic of the Journal. The article does not refer to sustainable development, but examines the behaviour of the parties to the land lease agreement.
- The article requires stylistic refinement - the language sometimes does not meet scientific standards.
- The authors use the term "land transfer" interchangeably with "land rent". Firstly, it would be more appropriate to use the phrase "transfer of rights", which consists of the transfer of ownership, usufruct, perpetual usufruct, tenancy, etc. Secondly, these are not synonyms.
- The authors have not clearly demonstrated why the land market in China is based on informal agreements.
- The introduction lacks an explanation of how the land market in China works, how it is regulated, etc.
- The theoretical chapter is not exhaustive, and the description of market-oriented economy suggests a lack of knowledge in this area.
- The construction and fragmentation of hypotheses, which could be successfully replaced by research questions, is surprising.
- Can the research carried out in Henan province be considered representative? The authors do not justify it.
- The research was conducted in 2014, so the presented trends may be out of date.
- The conclusions seem to be too local.

Reviewer 3 Report

The authors present the results of a survey of representative farmers in Henan Province, China, regarding the use of land contracts in land rentals. They note that traditionally land agreements were between family members and were largely oral in character; more complex agreements, especially with non-family members, are tending to be written. They note that there seems to be a trend toward more written agreements as such agreements become more complex and involve actors other than family members. This trend is moving agriculture onto a more formal basis. 

The name of the province should appear in the title and abstract, perhaps within parentheses on line 3 (title) and in the text on line 8 (abstract: "largest wheat-producing Henan Province").

On line 14 of the abstract and elsewhere in the discussion (line 451 for example), the term "civilization" should be either defined or replaced with a more descriptive term. 

The organization of the manuscript should be reviewed: much of the discussion is in the Introduction and in the Methods & Data rather than in the section Results & Analysis. The authors also seem to have used multiple referencing techniques: the reference list uses the numeric referencing system (each reference is numbered) but the text retains the Harvard (author, date) referencing system with periodic use of footnotes. The footnotes should be incorporated into the text; perhaps the conversion of mu to ha could be included in the text--e.g., 9.4 mu (0.6 ha) on line 35 for example. The National Rural Fixed Observation Point System on line 34 should be included in the reference list and the Ministry of Agriculture Planning Bureau (footnote 1) also should be included in the reference list. The Information Office of State Council, line 50 and footnote 3, should be included in the reference list.

On line 78, there is some missing text (even to make it available for free?). On line 186, it seems that "increasing" should actually be decreasing. On line 205, "change" should be charge.

Beginning on line 206 in the Methods & Data section the past tense should be adopted consistently throughout this section. In Table 1 and elsewhere, a better term for "dummies" (item (3) should be used; perhaps just Region would be appropriate. This section includes results which should be moved to the results section. Beginning on line 362, "labors" should be "laborers".

The references are appropriate and current.

This paper is recommended for publication after significant re-organization to better conform to the journal requirements. The English usage is reasonable but one further review would be beneficial to ensure consistency in tense.

Round 2

Reviewer 2 Report

First of all, I must admit that the Authors have put a lot of effort into improving the article. In its present form the article is much more understandable and valuable.

However, the choice of vocabulary needs to be improved:
- in line 6 - just use the term"land market"
- in line 8, the form 'land rental use right transfer contracts' is not correct
- alike on line 9

The correct terms are:

- land use rights
- rural/arable land use rights
- transfer of land use rights
- land lease rights
- land lease agreement (with lessor and lessee as parties) instead of land rental contract

These terms need to be improved throughout the whole text of the article.

However, the consistency of the content with the scope of the Journal is left to the editors' discretion.

Author Response

Responses to Reviewer 2

First, we would like to thank the Reviewer 2 for his/her thoughtful comments and suggestions to our manuscript and contribution to our revision. As usual, we first introduce the reviewer’s comments in italics and then followed by our responses.

First of all, I must admit that the Authors have put a lot of effort into improving the article. In its present form the article is much more understandable and valuable.

Thank you for the kind recognition of our paper and work.

Point 1: However, the choice of vocabulary needs to be improved:
- in line 6 - just use the term" land market"
- in line 8, the form 'land rental use right transfer contracts' is not correct
- alike on line 9

The correct terms are:
- land use rights
- rural/arable land use rights
- transfer of land use rights
- land lease rights
- land lease agreement (with lessor and lessee as parties) instead of land rental contract

These terms need to be improved throughout the whole text of the article.

Thank you for pointing out the right vocabulary of these terms in this manuscript. As suggested, we have refined the terms used throughout the article. Please see the revision as follows:

(Revision, lines 6, 64, 153, 158, 179, 181, 183)

“land rental/use right transfer market” → “land market”

(Revision, Lines 2-3, 8, 9, 10, 13, 14, 22,70, 77, 79, 82, 83, 84-85, 86, 90-91, 116, 122, 126, 131, 167-168, 192, 204, 207, 208, 218, 230, 394, 425-426, 439, 449-453, 473, 475-476, 493, 501-504, 527, 544, 546, 548, 557, 565, 568, 571, 582-583, 591-592,601, 609, 611, 612, 649, 653-654, 703, 705)

“land rental use right transfer contracts” → “land lease agreement(s)”

(Revision, Lines 14-15, 23, 51-52, 72, 87, 128-129, 132-133, 191, 206, 210, 216, 219, 220, 236, 242, 254, 480-482, 534, 551, 650, 662, 664-665, 668-669, 673-674, 676, 685, 697, 714)

“land use right” → “transfer of (rural) land use rights”

(Revision, lines 56, 60)

“land use right transfer” → “land transfer”

(Revision, lines 73, 75, 118-119, 120, 123, 134, 169, 173, 223, 232, 237, 242-243, 244, 252-253, 295, 309, 379, 479, 485, 487, 488, 491-492, 496-498, 513, 517, 529, 532, 537-538, 540, 555, 574-576, 578, 584, 587-589, 597, 604, 606, 609, 656, 664, 666, 671-672, 676, 677, 679-680, 682-684)

“(land rental) contract” → “land lease agreement(s)”

(Revision, line 101)

“rural land transfer” → “(the) transfer of rural land”

(Revision, line 145)

“National laws have deprived rural land of the right to reasonable flow” →“National laws have deprived the reasonable transfer of rural land use rights”

(Revision, lines 185, 186, 694)

“farmland use right” → “rural land use rights”

However, the consistency of the content with the scope of the Journal is left to the editors' discretion.

Thank you for your valuable comments on this article.

Reviewer 3 Report

The authors have significantly re-arranged the manuscript to better reflect the logic of the major sections, with results and discussion elements moved into that section of the paper. While attending to the major comments previously offered, the authors have substituted the term citizenization for civilization, but this reviewer remains somewhat confused by this terminology. it is understood that the idea is a movement toward a more formal legal arrangement, so perhaps using a few more words would better convey this idea. Also, since the footnotes have been incorporated into the text, they can be deleted from the foot of the respective pages.

This study is well worthy of publication as it documents the transition to a more formal land management structure within a society where land is held in common ownership--e.g., a transitional stage toward land ownership.

With the minor refinement of terminology, the paper would seem to be ready for publication.

Author Response

Responses to Reviewer 3

First, we would like to thank the Reviewer 3 for his/her thoughtful comments and suggestions to our manuscript and contribution to our revision. As usual, we first introduce the reviewer’s comments in italics and then followed by our responses.

Point 1: The authors have significantly re-arranged the manuscript to better reflect the logic of the major sections, with results and discussion elements moved into that section of the paper.

Thank you for the kind recognition of our paper and work.

Point 2: While attending to the major comments previously offered, the authors have substituted the term citizenization for civilization, but this reviewer remains somewhat confused by this terminology. It is understood that the idea is a movement toward a more formal legal arrangement, so perhaps using a few more words would better convey this idea.

Thanks for this comment. As suggested, we have made a brief explanation of this term. Please refer to lines 689-691 for this revision as follows:

(Revision, lines 689-691):

…citizenization (i.e., the process in which the agricultural population transfers from the countryside to the cities and gradually becomes urban residents, that is to say, the  farmers’ hukou changes from the countryside to the city).

Also, since the footnotes have been incorporated into the text, they can be deleted from the foot of the respective pages.

As suggested, we have deleted the footnotes in the clean version.

This study is well worthy of publication as it documents the transition to a more formal land management structure within a society where land is held in common ownership--e.g., a transitional stage toward land ownership. With the minor refinement of terminology, the paper would seem to be ready for publication.

Thank you for the kind recognition of our paper and work.